# Bariatric Surgery Induces Alterations in the Immune Profile of Peripheral Blood T Cells

**DOI:** 10.3390/biom14020219

**Published:** 2024-02-12

**Authors:** Pedro Barbosa, Aryane Pinho, André Lázaro, Diogo Paula, José G. Tralhão, Artur Paiva, Maria J. Pereira, Eugenia Carvalho, Paula Laranjeira

**Affiliations:** 1University of Coimbra, Institute for Interdisciplinary Research, Doctoral Programme in Experimental Biology and Biomedicine (PDBEB), 3030-789 Coimbra, Portugal; pedrobarbosa@cnc.uc.pt; 2Center for Neuroscience and Cell Biology (CNC), University of Coimbra, 3004-504 Coimbra, Portugal; aryanepinho@cnc.uc.pt; 3Institute for Interdisciplinary Research (IIIUC), University of Coimbra, 3030-789 Coimbra, Portugal; 4Center for Innovative Biomedicine and Biotechnology (CIBB), University of Coimbra, 3000-504 Coimbra, Portugal; artur.paiva@chuc.min-saude.pt; 5Department of Life Science, University of Coimbra, 3000-456 Coimbra, Portugal; 6General Surgery Unit, Centro Hospitalar e Universitário de Coimbra, 3000-075 Coimbra, Portugal; uc42311@uc.pt (A.L.); 17681@chuc.min-saude.pt (D.P.); jgtralhao@chuc.min-saude.pt (J.G.T.); 7Clinical Academic Center of Coimbra (CACC), 3004-061 Coimbra, Portugal; 8Coimbra Institute for Clinical and Biomedical Research (iCBR), Group of Environmental Genetics of Oncobiology (CIMAGO), Faculty of Medicine (FMUC), University of Coimbra, 3000-548 Coimbra, Portugal; 9Institute of Biophysics, Faculty of Medicine, University of Coimbra, 3000-548 Coimbra, Portugal; 10Flow Cytometry Unit, Department of Clinical Pathology, Centro Hospitalar e Universitário de Coimbra, 3000-076 Coimbra, Portugal; 11Instituto Politécnico de Coimbra, ESTESC-Coimbra Health School, Ciências Biomédicas Laboratoriais, 3046-854 Coimbra, Portugal; 12Department of Medical Sciences, Clinical Diabetology and Metabolism, Uppsala University, SE-75185 Uppsala, Sweden; maria.pereira@medsci.uu.se

**Keywords:** T cells, Treg, obesity, bariatric surgery, insulin resistance, immune phenotype

## Abstract

Low-grade inflammation is closely linked to obesity and obesity-related comorbidities; therefore, immune cells have become an important topic in obesity research. Here, we performed a deep phenotypic characterization of circulating T cells in people with obesity, using flow cytometry. Forty-one individuals with obesity (OB) and clinical criteria for bariatric surgery were enrolled in this study. We identified and quantified 44 different circulating T cell subsets and assessed their activation status and the expression of immune-checkpoint molecules, immediately before (T1) and 7–18 months after (T2) the bariatric surgery. Twelve age- and sex-matched healthy individuals (nOB) were also recruited. The OB participants showed higher leukocyte counts and a higher percentage of neutrophils. The percentage of circulating Th1 cells were negatively correlated to HbA1c and insulin levels. OB Th1 cells displayed a higher activation status and lower PD-1 expression. The percentage of Th17 and Th1/17 cells were increased in OB, whereas the CD4^+^ Tregs’ percentage was decreased. Interestingly, a higher proportion of OB CD4^+^ Tregs were polarized toward Th1- and Th1/17-like cells and expressed higher levels of CCR5. Bariatric surgery induced the recovery of CD4^+^ Treg cell levels and the expansion and activation of Tfh and B cells. Our results show alterations in the distribution and phenotype of circulating T cells from OB people, including activation markers and immune-checkpoint proteins, demonstrating that different metabolic profiles are associated to distinct immune profiles, and both are modulated by bariatric surgery.

## 1. Introduction

Obesity is a major health concern. It can trigger a number of diseases and conditions, including insulin resistance (IR), type 2 diabetes mellitus (T2D), and cardiovascular disease (CVD) [1,2]. Obesity-associated chronic low-grade inflammation is a common feature of these comorbidities [1]; therefore, the immune system has become a major focus of investigation of obesity [2,3].

Innate and adaptive immune cells play an important role in maintaining homeostasis and a perfect balance between anti-inflammatory and pro-inflammatory cell populations, including those infiltrated in the adipose tissue (AT) microenvironment. However, the adipokine-releasing patterns associated with dysfunctional adipocytes, present in obesity, prompts an imbalance within the AT-infiltrated immune cell populations. Increased production of interleukin (IL)-6, tumor necrosis factor (TNF)-α, and leptin by dysfunctional adipocytes not only modulates the local inflammation by induction of pro-inflammatory polarization of immune cells but also plays an important role in systemic inflammation [4,5,6]. Furthermore, a chronic pro-inflammatory profile of immune cells may have a deleterious impact on metabolism, contributing to insulin resistance and T2D development. Neutrophils and, to a higher extent, monocytes/macrophages have been described as major contributors to obesity-related low-grade inflammation and insulin resistance development [7,8,9]. 

Adaptive immune cells also play an important role in obesity-related inflammation. CD4^+^ (Th) and CD8^+^ (Tc) T lymphocytes have been described as displaying an altered phenotype and function in obesity and metabolic dysfunction [10,11]. In fact, studies suggest that T cells display a metabolic switch toward pro-inflammatory polarization, such as Th1 and Th17 [8,12,13]. Moreover, both the number and the percentage of circulating CD4^+^ Treg cells within CD4^+^ T cells are reduced in adults with morbid obesity [14]. In turn, important phenotypic alterations are present in CD8^+^ Treg cells from individuals with obesity (our unpublished data). And changes in the functional compartments of CD8^+^ T cells have also been described in people with morbid obesity and the metabolic syndrome [6,15]. Despite these findings, little is known about the CD8^+^ T cell phenotype, either in obesity or T2D. Moreover, the information available about less-represented T cell subsets is even scarcer.

In fact, most of the available literature focuses on a restricted type and number of T cell subsets, such as Th1, Th17, or Treg [8,14]. Thus, there is lack of information on less-represented T cell subsets, which may play an important role in obesity-associated low-grade inflammation. In addition, a deep evaluation of the circulating immune cells in obesity, covering most T cell subsets and providing the big picture of T cell changes in obesity, is missing. Therefore, the present study describes to a large extent the circulating T cells in people with obesity at different metabolic states. The absolute counts and percentages of 44 different T cell subsets, which are further characterized in terms of activation status and expression of immunosuppressive molecules, are assessed in this study, in participants with obesity and in a non-obese control group. Additionally, the impact of bariatric surgery on each of the circulating T cell subsets analyzed has also been studied.

## 2. Materials and Methods

### 2.1. Participants

A group of 41 adults with obesity (OB), comprising 27 women and 14 men with a mean age of 44 ± 10.8 years old, undergoing bariatric surgery at the Bariatric Surgery Unit, Department of General Surgery, Centro Hospitalar e Universitário de Coimbra (CHUC), Portugal, were enrolled in the present study, after giving their written informed consent. This study was approved by the CHUC’s Ethical Committee (protocol code: CHUC-115-20 and CHUC-202-20), and it was performed according to the Declaration of Helsinki (1964) guidelines. The participants were clinically indicated for bariatric surgery following specific criteria of BMI > 40 kg/m^2^ or ≥35 kg/m^2^ associated with dyslipidemia, hypertension, clinical diagnosis of T2D, or sleep apnea. Acute inflammatory conditions, cancer, and neurodegenerative and autoimmune diseases, as well as the use of immunosuppressive and anti-inflammatory drugs on a daily basis, were exclusion criteria. From the recruited participants, 35 were studied before bariatric surgery and 14 after bariatric surgery, as indicated in Figure 1 and Table 1. Eight of the participants were studied both before and after bariatric surgery, while six were studied exclusively after the bariatric surgery (Figure 1 and Appendix A). Only individuals with Class IV obesity undergo a second bariatric surgery. An age- and sex-matched group of healthy individuals (*n* = 12; 8 women and 4 men; mean age: 43 ± 11.9 years old) were also recruited to participate as a control group (non-obese group, nOB). A volume of 12 mL of EDTA-anticoagulated fasting peripheral blood (PB) was collected from all participants. 

#### 2.1.1. Anthropometric and Biochemical Characterization of the Participants

The anthropometric and biochemical/metabolic characteristics of the participants are detailed in Table 1. A scale (seca 515 mBCA, seca, Hamburg, Germany) was used to measure the body weight (kg). The body mass index (BMI, kg/m^2^) was calculated as previously described [16]: BMI = BW (kg)/height^2^ (m^2^). The waist (WC), hips (HC), and neck (NC) circumferences were measured using a flexible measuring tape (cm), while a digital sphygmomanometer (Philips IntelliVue MP20, Philips, Boebligen, Germany) was used to measure the systolic and diastolic blood pressure (mmHg) after 5 min of rest. 

Glycated hemoglobin (HbA1C, %), fasting glucose (mg/dL) and insulin (µU/mL), C-peptide (µg/mL), and C-reactive protein (CRP, µg/mL), as well as triglycerides (TG, mg/dL), high-density lipoprotein cholesterol (HDL, mg/dL), and total cholesterol (mg/dL) were measured in PB after an 8 h fast, following the standard methods at the Department of Clinical Pathology at CHUC. The Friedewald formula [17], LDL = TC − HDL + (TG × 0.2), was used to calculate the low-density lipoprotein cholesterol (LDL, mg/dL), while the homeostatic model assessment insulin resistance index (HOMA-IR) was also calculated using the fasting glucose and insulin levels as previously described [16]: HOMA-IR = (insulin (µU/mL)) × glucose (mg/dL))/405 (Table 1). 

Leptin and adiponectin levels were measured in plasma, by immunoassay, following the manufacturer’s instructions (RayBio Tech Life Inc., Peachtree Corners, GA, USA) (Table 1).

#### 2.1.2. Metabolic profile and obesity class of participants

Participants with obesity were divided per obesity class [18], into Class II (35 ≤ BMI < 40 kg/m^2^), Class III (40 ≤ BMI < 50 kg/m^2^), or Class IV (BMI ≥ 50 kg/m^2^), as well as according to their metabolic status [19], based on HOMA-IR, into insulin sensitive (IS) or IR. The IR group was subdivided into either normoglycemic (IRn), with pre-diabetes (Pre-T2D), or with type 2 diabetes (T2D), according to their fasting glucose and HbA1c % (Table 2).

Table 3 shows how participants from the different metabolic groups are distributed into obesity classes.

### 2.2. Immunophenotyping

PB immune cells were identified and characterized by flow cytometry, using a stain–lyse–wash procedure previously described [20,21,22]. In summary, 100 µL of PB was incubated with the monoclonal antibodies (mAbs) indicated in Table 4, in the presence of 50 µL of Brilliant stain buffer (Becton Dickinson Biosciences (BD), San Jose, CA, USA), for 30 min in the dark and at room temperature. Subsequently, erythrocytes were lysed using 2 mL of FACSLysing solution (BD) and a 10 min incubation period. After centrifugation at 540× *g* for 5 min, the FACSLysing solution was discarded, and the resulting cell pellet was washed with 2 mL of Dulbecco’s phosphate-buffered saline (DPBS; Corning, Manassa, VA, USA). Lastly, the cell pellet was resuspended in 500 µL of DPBS, and the sample was acquired in a FACSLyric^TM^ (BD) flow cytometer, using the FACSuite acquisition software (v1.5.0.925; BD). 

#### Identification and Characterization of the Major Immune Cell Populations and T Cell Subsets

Flow cytometry data were analyzed with the Infinicyt software (version 2.0.5; Cytognos SL, Salamanca, Spain). Appendix A describes the gate strategy used to identify the major immune cell populations (lymphocytes, monocytes, neutrophils, and eosinophils) and the distinct T cell subsets. Immune cell analysis was performed after excluding debris and cell doublets, considering their forward scatter (FSC)-A and FSC-H properties. Neutrophils were identified based on their FSC and side scatter (SSC) light dispersion characteristics, while monocytes were also identified based on CD4 expression (Figure 2A). T and B cells were identified by their positivity for CD3 (Appendix A) and CD20 (Appendix A), respectively. Our mAb combination (Table 4) was designed to perform a deep characterization of T lymphocytes, allowing us to identify 44 different subpopulations. Thus, T lymphocytes were primarily divided into CD4^+^ (Th), CD8^+^ (Tc), and CD4^+^CD8^+^ T cells (Appendix A). As CD4^−^CD8^−^ T cells comprised both TCRαβ CD4^−^CD8^−^ T cells and γδ T cells, our mAb combination did not allow for the distinction of these cell groups; therefore, they were not further analyzed. As demonstrated in Appendix A, within each one of the three major T cell subpopulations (Th, Tc, and CD4^+^CD8^+^ T cells), we identified the following T cell subsets: CD25^high^CD127^low/−^ regulatory T (Treg) cells (Appendix A); CXCR5^+^ follicular T (Tf) cells (Appendix A); and, based on CCR5 (CD195) and CCR6 (CD196) expression (Appendix A), CCR5^+^CCR6^−^ T cells (T1), CCR5^−^CCR6^+^ T cells (T17), CCR5^+^CCR6^+^ T cells (T1/17), and CCR5^−^CCR6^−^ T cells. Then, Treg cells were subdivided into CXCR5^+^ regulatory T (Tfr) cells (Appendix A). According to the expression of CCR5 and CCR6, T1-like, T17-like, T1/17-like, and CCR5^−^CCR6^−^ T cells were further identified within Treg, Tf, and Tfr cells. Finally, the percentage of early activated T cells (CD25^+^, Appendix A) and the expression of the immune-checkpoint proteins PD-1 (CD279, Appendix A) and TIM-3 (CD366, Appendix A) were further identified within each one of the 44 T cell subsets identified in this study, whenever the number of events acquired was enough to enable this analysis.

### 2.3. Statistical Analysis

Data were presented using mean ± standard deviation or median and inter-quartile range (iqr) for all studied variables. The comparison between groups was performed using the Wilcoxon rank sum and Kruskal–Wallis test from package *coin* version 1.4.2 [23,24], followed by Dunn’s test from package *rstatix* version 0.6.0 [25], as appropriate. Furthermore, for the participants studied both before and after bariatric surgery, paired-sample comparisons were performed using the Wilcoxon signed-rank test [26]. Spearman’s correlations were performed between two continuous variables, using the rcorr() function from package *Hmisc* version 4.4-2 [27]. Statistically significant differences were considered when *p* < 0.05. The plots were performed using the package *ggplot2* [28]. All statistical analysis was performed using R (version 4.0.2; R Foundation for Statistical Computing, Vienna, Austria) [26,29]. 

## 3. Results

### 3.1. Obesity Induces Alterations in Peripheral Blood Neutrophils and Monocytes

People with obesity (OB) presented higher white blood cell (WBC) counts (7.6 cells/µL ± 2.2) in comparison to the nOB group (6.2 × 10^3^ cells/µL ± 1.5, *p* < 0.05; Figure 2A and Appendix A). Additionally, the stratification of BMI into obesity classes indicated that Class II obesity (6.0 × 10^3^ cells/µL ± 2.1) displayed a similar WBC count to the nOB group. On the other hand, Classes III (7.9 x10^3^ cells/µL ± 1.7, *p* < 0.05 vs. nOB) and IV (8.2 × 10^3^ cells/µL ± 2.4, IV vs. II and IV vs. nOB, *p* < 0.05) were significantly increased (Figure 2A and Appendix A). Furthermore, the BMI was positively correlated to WBC (rho = 0.39, *p* < 0.05), as shown in Figure 3. A 1.4-fold increase in the percentage of neutrophils in the OB group compared to nOB (*p* < 0.05, Figure 2B and Appendix A) was also detected, which translated into an increase of 1.7× the absolute number of neutrophils (Appendix A). Additionally, the BMI and neck circumference were highly correlated to the percentage of neutrophils (rho = 0.62 and rho = 0.58, respectively, *p* < 0.05, Figure 3). Monocytes from the OB group showed a higher percentage of CCR5^+^ cells (Figure 2D and Appendix A) and a positive correlation between the percentage of monocytes and the HbA1c % for the OB group (rho = 0.36, *p* < 0.05; Figure 3). Considering the metabolic profile, the landscape was quite different. The IS group displayed a similar WBC count as the nOB group (nOB: 6.2 cells/µL ± 1.5; IS: 6.1 cells/µL ± 1.8; Figure 2A and Appendix A), while the non-IS OB groups showed higher WBC counts, with Pre-T2D (Pre: 8.8 cells/µL ± 3.0) displaying a significant difference vs. the nOB group (Figure 2A and Appendix A).

### 3.2. Obesity Alters the T Cell Percentage and Their Subsets, but Not That of B Cells, in Peripheral Blood

The percentage of lymphocytes in whole blood was decreased by 1.5-fold in the OB group when compared to nOB (*p* < 0.05; Figure 4A and Appendix A) and was negatively correlated to BMI and neck circumference (rho = −0.53 and rho = −0.44, respectively, *p* < 0.05), as displayed in Figure 5. Similarly, the percentage of T lymphocytes in whole blood was also decreased in the OB group (19% ± 6.8 vs. nOB: 26% ± 7.8, *p* < 0.05, Figure 4B and Appendix A) and negatively correlated to BMI and neck circumference (rho = −0.34 and rho = −0.46, respectively, *p* < 0.05; Figure 5). Importantly, the levels of plasma glucose showed a weak negative correlation to the percentage of lymphocytes and T lymphocytes (rho = −0.35 and rho = −0.31, respectively, *p* < 0.05; Figure 5). B lymphocytes did not show any differences among the studied groups (Figure 4C and Appendix A).

A significant increase in the percentage of CD4^+^ T cells in the OB group (65% ± 12.0) vs. nOB (54% ± 7.1, *p* < 0.05; Figure 4D and Appendix A) was observed, while the percentage of CD8^+^ T cells was significantly reduced (nOB: 38% ± 8.6 vs. OB: 30% ± 11.1, *p* < 0.05; Figure 4E and Appendix A). No differences were detected on CD4^+^CD8^+^ T cells (Figure 4F and Appendix A). BMI showed a moderate positive correlation with the percentage of CD4^+^ T cells (rho = 0.30, *p* < 0.05) (Figure 5). 

Looking at the metabolic profile, Pre-T2D (68% ± 5.8) and T2D groups (66% ±13.4) showed the highest percentage of CD4^+^ T cells in comparison to the nOB (54% ± 7.1, *p* < 0.05) group (Figure 4D and Appendix A). Moreover, the CD8^+^ T cells were decreased in insulin-resistant groups, with an emphasis in the Pre-T2D group (27% ± 6.1) when compared to nOB (38% ± 8.6, *p* < 0.05), as described in Appendix A and Figure 4E.

### 3.3. CD4^+^ T Lymphocytes

The phenotype of the CD4^+^ T cells, as well as the percentage of activated T cells (identified as CD25^+^), and the expression of immune regulatory molecules were summarized in Figure 6, Figure 7, Figure 8 and Figure 9, and more details are provided in Appendix A.

#### 3.3.1. Th1 Cells from OB Participants Present a Higher Activation Status and Lower Expression of PD-1

Though no differences in the percentage of Th1 cells between nOB and OB were found (Figure 6A and Appendix A), Th1 cells from OB displayed an increased percentage of activated cells (30% ± 17.2 vs. 19% ± 8.9 in nOB), more marked in Class III (39% ± 16.5, *p* < 0.05) and not verified in Class II (16% ± 6.4), as indicated in Figure 6E and Appendix A. In turn, a decrease in the percentage of PD-1^+^ Th1 cells was found among OB subjects from Class III (65% ± 15.4, *p* < 0.05, Figure 6G and Appendix A)) and Class IV (66% ± 20.0, *p* > 0.05) in comparison to nOB (71% ± 27.1), as seen in Figure 6G and Appendix A. Interestingly, when evaluating the metabolic groups, a sequential increase in PD-1^+^ Th1 cells from IS (54% ± 19.8) to IRn (62% ± 16.7), Pre-T2D (67% ± 14.6), and T2D (75% ± 14.9; Appendix A) was observed. In addition, T2D patients also presented an increased expression in PD-1 (measured as mean fluorescence intensity, MFI), as compared to the nOB (*p* > 0.05), IS, and IRn (*p* < 0.05) groups, as shown in Appendix A. Importantly, besides presenting a lower percentage of PD-1^+^ Th1 cells, IS participants also showed a decreased percentage of TIM-3^+^ Th1 cells (2.3% ± 1.6) compared to nOB (4.2% ± 2.0, *p* > 0.05) and the remaining groups (Figure 6H and Appendix A). Furthermore, a negative correlation between the percentage of Th1 and the percentage of HbA1c, insulin, or c-peptide levels were observed in IS, IRn, and Pre-T2D participants. Additionally, the neck circumference was positively correlated to the expression of PD-1 by Th1 cells (rho = 0.46, *p* < 0.05; Figure 7).

#### 3.3.2. The OB Group Displays an Increased Percentage of Th17 and Th1/17 Cells That Is Associated with a Reduced Expression of TIM-3

The percentages of Th17 and Th1/17 cells from the OB group were increased (17% ± 5.2 and 10% ± 5.2, respectively) in comparison to the nOB (11% ± 4.1 and 5.2% ± 2.2, respectively, *p* < 0.05; Figure 6B,C and Appendix A). Interestingly, a decreased percentage of PD-1^+^ cells within the Th17 subset was detected in the OB group (37% ± 9.9), with an emphasis on Class III obesity (34% ± 6.5), compared to nOB (44% ± 15.4, *p* > 0.05; Appendix A), in addition to the Th1/17 cells (nOB: 64% ± 17.6 vs. OB: 58% ± 11.5, *p* > 0.05; Appendix A). Furthermore, the expression of PD-1 in both Th17 and Th1/17 cells were reduced in Class III, in comparison to nOB (*p* < 0.05, Appendix A). Similarly, there was a decreased percentage of TIM-3^+^ Th1/17 cells in the OB group (3.2% ± 2.2) compared to nOB (4.0% ± 3.1), more pronounced in Class IV (2.3% ± 0.8, *p* < 0.05 vs. nOB; Figure 6I and Appendix A). The TIM-3 MFI in Th17 and Th1/17 cells, also showed a tendency to decrease in OB vs. nOB (*p* > 0.05; Appendix A).

#### 3.3.3. CCR5^−^CCR6^−^ Helper T Cells Show Decreased Percentage and an Altered Phenotype in Insulin Resistant Groups

CCR5^−^CCR6^−^ Th cells comprised approximately 52% ± 9.0 of the CD4^+^ T cells in the nOB group, while in OB groups they comprised 39% ± 9.8 of the CD4^+^ T cell compartment (*p* < 0.05), even among obesity classes and metabolic groups (Figure 6D and Appendix A). It is noteworthy that, despite their reduced percentage, these cells were more activated in OB (37% ± 11.4 vs. nOB: 26% ± 10.2, *p* < 0.05), with Pre-T2D (40% ± 15.0) and T2D (39% ± 10.3) showing the highest percentages of activated cells (Figure 6F and Appendix A). This increased cell activation was accompanied by a reduction in the expression of TIM-3 on these cells in OB (487 ± 58 vs. nOB: 736 ± 534, *p* < 0.05), with Classes III and IV being the lowest (*p* < 0.05; Appendix A). Additionally, the IRn group also displayed a decrease in the expression of TIM-3 (471 ± 70.4) when compared to nOB and IS (*p* < 0.05), as shown in Appendix A. Moreover, there was a negative correlation between the BMI and the percentage of CCR5^−^CCR6^−^ Th cells (rho = 0.40, *p* < 0.05) and the expression of TIM-3 on CCR5^−^CCR6^−^ Th cells (rho = −0.32, *p* < 0.05; Figure 7).

#### 3.3.4. Regulatory Helper T Cells

Participants with obesity presented a decrease in the percentage of CD4^+^ regulatory T cells measured in whole blood (OB: 0.67% ± 0.3 vs. nOB: 0.89% ± 0.3, *p* < 0.05, Figure 8B). Interestingly, the higher the BMI, the lower the CD4^+^ Treg percentage, with Class IV obesity showing the lowest value (nOB: 0.89% ± 0.3; II: 0.69 ± 0.36; III: 0.70 ± 0.27; IV: 0.63% ± 0.9; nOB vs. IV, *p* < 0.05; Figure 8B). Additionally, the percentage of regulatory T cells measured in whole blood was also reduced in T2D (IS: 0.62 ± 0.22; IRn: 0.70 ± 0.36; Pre-T2D: 0.67 ± 0.31; T2D: 0.66 ± 0.28; T2D vs. nOB *p* < 0.05; Figure 8B).

A deeper analysis of the CD4^+^ Treg compartments showed a higher percentage of Th1-like and Th1/17-like Tregs in the OB group (6.5% ± 2.6 and 29% ± 10.9, respectively) vs. nOB (4.9% ± 2.3 and 22% ± 7.3, respectively, *p* < 0.05; Figure 8C,E and Appendix A). This was accompanied by a higher expression of CCR5 in both subpopulations (*p* < 0.05; Appendix A). These differences were more marked for Class III obesity (Th1-like: 7.3% ± 2.6; Th1/17-like: 29% ± 9.5, *p* < 0.05 vs. nOB) and IRn, which showed the highest percentage of Th1-like Treg (7.9% ± 3.3) (Figure 8C,E and Appendix A). An increased expression of CCR5 was observed in all IR groups, especially the T2D (*p* < 0.05, Appendix A). Moreover, the percentage of CCR5^−^CCR6^−^ Th-like Treg was significantly reduced in OB vs. nOB (*p* < 0.05; Figure 8F and Appendix A). Noteworthy, the percentage of PD-1^+^ Th1-like Treg was decreased in Pre-T2D (41% ± 8.1 vs. nOB: 56% ± 12.3; Figure 8H and Appendix A), while the PD-1^+^ Th1/17-like was reduced in the Pre-T2D group (36% ± 10.1, *p* < 0.05 vs. nOB; Figure 8M and Appendix A) and Class II obesity (39% ± 13.9), in comparison to the nOB (49% ± 13.3), with sequential increases in Classes III and IV (Appendix A). On the other hand, the percentages of TIM-3^+^ Th1-like and TIM-3^+^ Th1/17-like Tregs were higher in the T2D (Figure 8I,N and Appendix A).

No differences were found in the percentage of Th17-like Treg between OB and nOB groups. However, the IRn presented a reduction in the percentage of Th17-like Tregs (16% ± 8.2) when compared to the nOB group (25% ± 10.1, *p* < 0.05), as shown in Figure 8D and Appendix A. 

Interestingly, when looking at the percentages of PD-1^+^ Th17-like Tregs, these were reduced in Pre-T2D in comparison to the nOB and the remaining OB groups (Figure 8J and Appendix A). On the other hand, the IRn showed a higher expression of PD-1 by Th17-like Treg as compared to nOB, Pre-T2D, and T2D (*p* < 0.05). The percentage of TIM-3^+^ Th17-like Treg was increased in the Class II and Pre-T2D groups, when compared to nOB (*p* > 0.05), and it was decreased in the IS group (*p* > 0.05), as seen in Figure 8L and Appendix A. Of note, the IS group showed a similar distribution in the Treg phenotypes as the nOB group, except in the percentage of CCR5^−^CCR6^−^ Th-like Treg, which was significantly reduced (nOB: 18% ± 5.6 vs. IS: 10% ± 4.5, *p* < 0.05; Appendix A).

Importantly, almost 31% of the CD4^+^ Tregs in circulation, from people with obesity, showed a Tfr phenotype, while 29% presented a Th1/Th17-like phenotype (Figure 9A and Appendix A). Similar to Th1-like Treg, CD4^+^ Tfr showed a higher percentage of cells with a Th1-like phenotype within Class III obesity and IRn, when compared to nOB (*p* < 0.05; Figure 9B and Appendix A). Interestingly, the percentage of PD-1^+^ Th1-like Tfr was increased in Class II (64% ± 19.4) and IS (71% ± 14.4) compared to nOB (*p* < 0.05; Figure 9C and Appendix A). Furthermore, obesity Class II patients also had higher percentage of TIM-3^+^ Th1-like CD4^+^Tfr cells (11% ± 4.6) compared to nOB (*p* < 0.05) (Figure 9D and Appendix A).

#### 3.3.5. Follicular Helper T Cells

Tfh cells play an important role in B cell differentiation, and our results pointed to a slight increase in the percentage of these cells in people with obesity (nOB: 15% ± 4.3 vs. OB: 18% ± 4.0, *p* < 0.05), with no differences within the groups with obesity (Figure 10A and Appendix A). Interestingly, despite this increase in Tfh, no changes were observed in B lymphocytes. Furthermore, the expression of CXCR5 (Appendix A) was increased in the Tfh from OB participants (3224 ± 748 vs. nOB: 2766 ± 373, *p* < 0.05), especially in Pre-T2D (3501 ± 847, *p* < 0.05 vs. nOB). Interestingly, the percentage of Th1/17-like Tfh was higher in people with IRn (3.8% ± 1.3) compared to nOB (2.4% ± 1.5, *p* < 0.05; Appendix A). The percentage of PD-1^+^ Tfh was reduced in people with Class III (44% ± 8.4) obesity compared to the remaining OB and nOB groups (54% ± 9.9, *p* < 0.05; Figure 11B and Appendix A). Likewise, the percentage of TIM-3^+^ Tfh cells was decreased in Class IV (2.0% ± 0.73, *p* > 0.05) and IRn (1.9% ± 0.93, *p* < 0.05) compared to nOB (2.9% ± 1.2) (Figure 11C and Appendix A). The percentage of Th1/17-like Tfh in nT2D was negatively correlated to HbA1c (rho = −0.44, *p* < 0.05) and glucose (rho = −0.51, *p* < 0.05) (Figure 7).

### 3.4. CD8^+^ T Lymphocytes

The phenotypic features of CD8^+^ T cells, the percentage of activated cells (CD25^+^), and the expression of immune regulatory proteins were summarized in Figure 11, Figure 12 and Figure 13, and more details are provided in Appendix A.

#### 3.4.1. Tc1 Cells Display Higher Expression of CCR5 in the OB Group and It Was Correlated to Insulin Levels

In similarity to the Th1 cells, there were no differences in the percentages of Tc1 among groups (Figure 11A and Appendix A). However, OB participants presented higher expression of CCR5 (measured as MFI) in Tc1 cells (3703 ± 1853), compared to nOB (2646 ± 1091, *p* < 0.05), particularly in the IS group (4164 ± 1671, *p* < 0.05) and T2D (3884 ± 2039, *p* < 0.05), as seen in Appendix A. Additionally, the percentage of PD-1^+^ Tc1 cells was decreased in the IS (27% ± 11.4), while the remaining groups showed similar percentages to the nOB (41% ± 17.3, *p* > 0.05; Figure 11E and Appendix A). Interestingly, the percentage of PD-1^+^ Tc1 cells was highly correlated to insulin levels and HOMA-IR (rho= 0.50 and rho = 0.51, *p* < 0.05, respectively; Figure 12) for the nOB and nT2D groups.

#### 3.4.2. In Tc17 Cells, the PD-1 Expression Correlates to Metabolic Parameters

An increase in the percentage of Tc17 was observed in the OB group (OB: 2.1% ± 1.4 vs. nOB: 1.3% ± 0.59, *p* > 0.05; Figure 12 and Appendix A). This was accompanied by a higher expression of CCR6 measured as MIF (OB: 983 ± 245 vs. nOB: 790 ± 206, *p* < 0.05) (Appendix A). On the other hand, nT2D showed a higher percentage of Tc17 (2.33% ± 1.6) compared to nOB (1.3 % ± 0.59, *p* < 0.05), with the Pre-T2D group presenting the highest expression CCR6 (*p* < 0.05 vs. nOB) (Figure 11B and Appendix A). Interestingly, Tc17 cells from nT2D participants presented a significant correlation between the expression of PD-1 (measured as MFI) and insulin (rho = −0.47), HOMA-IR (rho = −0.48), and C-peptide (rho = −0.49), as represented in Figure 6.

#### 3.4.3. Increased Activation Status of Cytotoxic Tc1/17 and CCR5^−^CCR6^−^ T Cells in the OB Group

The percentage of Tc1/17 cells was similar among groups (Figure 11C and Appendix A); however, they displayed increased activation status in the OB (24% ± 14.1 vs. nOB: 17% ± 11.6, *p* < 0.05), which was more pronounced in Class II obesity (32% ± 16.9, *p* < 0.05 vs. nOB) (Appendix A). It is noteworthy that the percentage of PD-1^+^ Tc1/17 cells tended to be decreased in OB (51% ± 11.7), especially in the Class III (45% ± 10.3), IS (49% ± 14.6) and IRn (48% ± 8.9) groups, compared to the nOB (57% ± 18.8, *p* > 0.05), (Figure 11F and Appendix A). Adiponectin levels were positively correlated to the percentage of TIM-3^+^ Tc1/17 cells (rho = 0.39, *p* < 0.05; Figure 12). Interestingly, CCR5^−^CCR6^−^CD8^+^ T cells also showed an activated pattern in the OB (11% ± 6.6) compared to the nOB (8.0% ± 8.3, *p* > 0.05), especially in Class II obesity (15% ± 6.7, *p* < 0.05) and Pre-T2D (15% ± 9.7, *p* < 0.05; Figure 11G and Appendix A).

#### 3.4.4. CD8^+^ Regulatory T Cells Are Expanded in Participants with Obesity

A deep characterization of CD8^+^ Treg cells has been described in detail in another study [30]. Briefly, an increase in the percentage of CD8^+^ Treg cells in the OB group (nOB: 0.14% ± 0.18 vs. OB: 0.27% ± 0.24, *p* < 0.05) was observed. This was more pronounced in pre-diabetes (Pre-T2D: 0.42% ± 0.31, *p* < 0.05 vs. nOB). Additionally, positive correlations between the percentage of circulating CD8^+^ Treg cells and fasting insulin (*p* < 0.05) or CRP (*p* < 0.05) levels [30] were found. Moreover, the CD8^+^ Treg phenotype from OB individuals were more prone to differentiate into Tc1- and Tc1/17-like cells and displayed increased expression of CCR5 [30]. 

#### 3.4.5. Follicular Cytotoxic T Cells Present an Increased Activation Status in the OB Group

A slight increase in the percentage of Tfc was detected in the OB (2.6% ± 1.8 vs. nOB: 1.9% ± 0.88, *p* > 0.05), particularly in the T2D (3.1% ± 2.0, *p* = 0.054 vs. nOB) (Figure 13A and Appendix A). In similarity to the Tfh, the Tfc also display higher expression of CXCR5 in OB (*p* < 0.05; Appendix A), particularly in Class II obesity, as well as a higher percentage of activated Tfc cells (OB: 10% ± 5.3 vs. nOB: 6.6% ± 4.0, *p* < 0.05; Figure 13B and Appendix A). 

Furthermore, the T2D group displayed an increase in Tc1-like Tfc in comparison to the nT2D (T2D: 69% ± 8.1 vs. nT2D: 60% ± 15.1, *p* < 0.05; nOB: 63% ± 12.8). On the other hand, the percentages of Tc1/17-like Tfc were decreased in Class III obesity (4.6% ± 2.3 vs. nOB: 8.2% ± 5.0, *p* < 0.05) and in the IRn (4.7% ± 3.4, *p* > 0.05 vs. nOB; Appendix A). Moreover, the Tfc from Class III obesity presented a reduction in the percentage and expression of PD-1, when compared to the nOB and the remaining obesity groups (Figure 13C and Appendix A). Importantly, the percentage of Tc17-like Tfc cells was positively correlated to leptin levels (rho = 0.47, *p* < 0.05), while the expression of PD-1 (per MFI) in B cells was negatively correlated to the percentage of CD25^+^ Tfc cells (rho = −0.43, *p* < 0.05) (Figure 12).

### 3.5. The Phenotype of CD4^+^CD8^+^ T Cells in Obesity

The CD4^+^CD8^+^ T cells comprised only 2% of the circulating T cells in both the nOB and OB groups (Figure 5 and Appendix A). In participants with OB, the CD4^+^CD8^+^ T cells did not exhibit any differences in their polarization compared to the nOB (Figure 14 and Appendix A). However, a decrease in T1/17-like CD4^+^CD8^+^ T cells was observed in Class II obesity (*p* < 0.05 vs. nOB; Figure 14C and Appendix A). Hence, the OB group displayed a higher percentage of activated T1/17-like CD4^+^CD8^+^ T cells (48% ± 17.2), especially in Class II (55% ± 15.6), compared to the nOB group (37% ± 15.6, *p* < 0.05) (Appendix A). A decreased percentage of PD-1^+^ T1-like CD4^+^CD8^+^ T cells in OB (57% ± 24.9 vs. nOB: 68% ± 26.2, *p* > 0.05; Appendix A) was also observed, especially in Class III obesity (50% ± 24.3). Likewise, a reduction in PD-1^+^ among T17-like and T1/17-like CD4^+^CD8^+^ T cells in the OB vs. nOB group (*p* < 0.05), more marked in Class III obesity, IRn, and Pre-T2D for the T17-like cells (*p* < 0.05) and in Class IV for the T1/17-like CD4^+^CD8^+^ T cells (*p* < 0.05), was observed (Appendix A). The OB participants had a lower percentage of TIM-3^+^ T1-like CD4^+^CD8^+^ T cells, especially in Class III obesity, IS, and IRn, compared to nOB (*p* < 0.05) (Appendix A). Interestingly, the percentage of TIM-3^+^ T1/17-like CD4^+^CD8^+^ T cells was negatively correlated to HbA1c levels in the nT2D group (rho = −0.54, *p* < 0.05; Figure 15). 

Looking deeper into the CCR5^−^CCR6^−^CD4^+^CD8^+^ T cells, they were markedly more activated in individuals with obesity (nOB: 17% ± 15.5 vs. OB: 30% ± 19.1, *p* < 0.05), especially Pre-T2D and T2D (*p* < 0.05 vs. nOB; Appendix A), and displayed a decreased percentage of PD-1^+^ cells (nOB: 48% ± 28.5 vs. OB: 31% ± 17.1, *p* = 0.085; Appendix A). This decrease was more pronounced in Class III obesity (22% ± 12.0, *p* < 0.05 vs. nOB) and Irn (27% ± 19.2, *p* = 0.069 vs. nOB; Appendix A). In addition, the percentage of activated CCR5^−^CCR6^−^CD4^+^CD8^+^ T was positively correlated to BMI (rho = 0.34, *p* < 0.05; Figure 15). 

The CD4^+^CD8^+^ Treg cells were decreased only in Class II obesity (0.34% ± 0.17 vs. nOB: 1.1% ± 1.1, *p* < 0.05; Figure 14E and Appendix A).

Furthermore, the Tf compartment was increased in T2D (3.1% ± 1.7) in comparison to nOB (2.3% ± 0.96, *p* > 0.05), IS (1.3% ± 0.50, *p* < 0.05) and IRn (1.7% ± 1.0, *p* < 0.05; Figure 14F and Appendix A). Moreover, participants with Class II obesity displayed an increase in the percentage of T17-like CD4^+^CD8^+^ Tf cells compared to nOB, while in the remaining groups, T17-like cells tended to be decreased (nOB: 15% ± 14.5; OBII: 20% ± 12.3; OBIII: 11% ± 9.5; OBIV: 6.9% ± 3.8; OBII vs. OBIV, *p* < 0.05; Appendix A). In similarity to T17-like cells, T1/17-like CD4^+^CD8^+^ Tf cells were also reduced in participants with Class III obesity (nOB: 10 ± 9.1, OBII: 13% ± 7.7, OBIII: 6.9% ± 5.1, OBIV: 12% ± 5.4; OBIII vs. OBIV, *p* < 0.05) and in the IRn group (6.8% ± 5.7, *p* > 0.05 vs. nOB; Appendix A). Interestingly, the IRn and Pre-T2D also displayed a slight reduction in the percentage of the PD-1^+^ CD4^+^CD8^+^ Tf cells, compared to the nOB group (*p* > 0.05; Appendix A).

### 3.6. Differences within Metabolic Groups

Participants with obesity and insulin resistance, not taking medication for T2D, showed a higher percentage of CD4^+^CD8^+^ T cells within T cells (IRn: 2.8% ± 1.1; Pre-T2D: 2.2% ± 2.0) compared to T2D patients under treatment (1.6% ± 1.9, *p* < 0.05 for IRn vs. T2D; Figure 5 and Appendix A). Furthermore, insulin-resistant groups (including T2D) displayed a higher polarization pattern toward the Th1-like, namely in CD4^+^ Treg cells (Figure 8 and Appendix A) and CD4^+^ Tfr cells (Figure 9 and Appendix A), when compared to the IS group. Importantly, groups with higher percentages of HbA1c displayed higher percentages of CD4^+^CD8^+^ Tf cells, while the polarization of these cells presented different patterns among the groups, wherein Pre-T2D displayed the highest percentage of CD4^+^CD8^+^ Tf toward T17-like (Appendix A). Additionally, the Pre-T2D also displayed the highest percentage of CD8^+^ Treg cells among groups [30]. 

Interestingly, the IS group showed an overall reduction in the percentage of cells expressing PD-1, as well as in the amount of PD-1 protein expressed per cell (measured as MFI), while a sequential increase was observed with increasing IR (Appendix A). A similar pattern was observed regarding the percentage of cells expressing TIM-3. These variations were especially observed in Th1 (Figure 6H and Appendix A) and Tc1 (Appendix A), as well as CD4^+^ Treg cells regarding TIM-3 (Figure 9G and Appendix A). On the other hand, the IS group had a higher percentage of PD-1^+^ Th1-like Treg cells (60% ± 8.7), while the Pre-T2D had the lowest percentage (41% ± 8.1, *p* < 0.05; Figure 8H and Appendix A) of these cells. As previously observed, the Pre-T2D group had important alterations in the CD8^+^ Treg cells, regarding their phenotype and the expression of immune checkpoint molecules [30].

### 3.7. Effect of Bariatric Surgery on Metabolic and Immune Profiles

To understand the effect of bariatric surgery on the obesity-associated immune profile, an unpaired analysis was performed, comparing 15 individuals with Class IV obesity before surgery (at baseline or T1) and a set of 14 participants, evaluated 7 to 18 months post-surgery (T2). Additionally, a follow-up study with paired comparisons including eight individuals with Class IV obesity, analyzed both at T1 and T2, was also performed (Figure 1). In this follow-up study, the time period between T1 and T2 ranged from 9 to 18 months (Figure 16 and Appendix A).

When comparing independent groups of patients at T1 and T2 (Figure 16 and Appendix A), a reduction in white blood cell count was detected after surgery (from 8.2 cells/µL ± 2.4 to 6.7 cells/µL ± 1.6, *p* < 0.05; Figure 16A and Appendix A), achieving values resembling those measured in the nOB group (6.2 cells/µL ± 1.5). Despite this reduction, the percentage of neutrophils remained similar before and after surgery (Appendix A). The percentage of lymphocytes was slightly increased at T2, despite remaining lower than in nOb (T1: 24% ± 7.8; T2: 29% ± 8.9; nOB: 40% ± 6.6; T2 vs. nOB, *p* < 0.05; Figure 16B and Appendix A). The percentage of T lymphocytes followed a similar pattern (Figure 16C and Appendix A). Interestingly, bariatric surgery impacted B lymphocytes, promoting increases in their percentage (T1: 2.1% ± 1.2; T2: 3.4% ± 1.7, *p* < 0.05; Figure 16E and Appendix A) and activation status (*p* < 0.05; Figure 16F), accompanied by an expansion of the Tfh cells (nOB: 15% ± 4.3; T1: 17% ± 4.9; T2: 20% ± 5.1; nOB vs. T2, *p* < 0.05) (Figure 16G and Appendix A). 

Regarding CD4^+^ T cells, the group participants evaluated post-surgery displayed an increased percentage of CD4^+^ Treg cells in whole blood (0.80% ± 0.34) compared to the group evaluated at baseline (0.63% ± 0.29, *p* > 0.05; Figure 16D). This increase was accompanied by an increased percentage of CD4^+^ Treg expressing PD-1 post-surgery (T2: 44% ± 11.0 vs. T1: 39% ± 11.02, *p* > 0.05; and vs. nOB: 33% ± 11.0, *p* < 0.05; Figure 16J and Appendix A), especially marked in Th1-like Tregs (T2: 62% ± 9.4 vs. T1: 54% ± 11.0, *p* < 0.05) (Figure 16K and Appendix A). An increased percentage of activated Th1 cells was detected post-surgery (36% ± 15.4) when compared to nOB (19% ± 8.9, *p* < 0.05) and at baseline (28% ± 17.4, *p* > 0.05; Figure 16H and Appendix A), and a similar activation pattern occurred in CCR5^−^CCR6^−^CD4^+^ T cells (Appendix A). 

Bariatric surgery also resulted in a slight increase in the percentage of PD-1^+^ CD8^+^ T cells (Figure 16O and Appendix A). In turn, T17-like CD4^+^CD8^+^ Tf cells were significantly increased at T2 (18% ± 18.6) vs. T1 (6.9% ± 3.8, *p* < 0.05) (Appendix A).

In paired comparisons of participants before (T1) and after (T2) bariatric surgery, an amelioration of the metabolic profile was observed, with the normalization of metabolic parameters, including HbA1c, glucose, insulin, HOMA-IR, and triglycerides. When the immune profile of these participants was analyzed (Appendix A), bariatric surgery induced a reduction in the WBC count (T1: 8.3 cells/µL ± 1.7 vs. T2: 6.7 cells/µL ± 1.5, *p* < 0.05; Figure 16A and Appendix A) and an increase in the percentage of lymphocytes (*p* < 0.05; Figure 16B and Appendix A), due to the increase in T (*p* < 0.05; Figure 16C and Appendix A) and B cells (*p* < 0.05; Figure 16E and Appendix A). Although no differences were found in the proportion of CD4^+^ Treg cells within CD4^+^ T cells (Appendix A), the percentage of CD4^+^ Treg measured in whole blood was increased post-surgery (T1: 0.58% ±0.27 vs. T2: 0.81% ± 0.31, *p* < 0.05; Figure 16D and Appendix A), along with an increased percentage of PD-1^+^ Th1-like Tregs (*p* < 0.05; Figure 16K and Appendix A) and the expression of CCR6 in Th17-like Tregs (*p* < 0.05; Appendix A). Similarly, CD4^+^ Tfr and Tfh cells were increased post-surgery (*p* < 0.05; Figure 16I and Appendix A), while the percentage of CCR5^−^CCR6^−^ Th cells was reduced (*p* < 0.05). Within CD8^+^ T cells, an increased percentage of Tc1/17 cells (Figure 16M and Appendix A), Tc1/17-like Tfc cells (Figure 16N and Appendix A), and PD-1^+^ Tfc cells was observed post-surgery (*p* < 0.05; Figure 16P and Appendix A). Conversely, the percentage of TIM-3^+^ cells was decreased among CD4^+^ T (*p* > 0.05) and CD8^+^ T cells (*p* < 0.05; Figure 16Q and Appendix A). Importantly, these decreases affected specific T cell subsets, namely the Tfh, Th1-like Treg, CCR5^−^CCR6^−^ Th, and CCR5^−^CCR6^−^ Tc cells (Appendix A).

## 4. Discussion

Obesity is a potential trigger for different comorbidities, with chronic low-grade inflammation perpetuating and underlying these conditions [1,2]. The immune system is paramount in stimulating and maintaining inflammatory processes. It plays an important role in the onset of obesity-related comorbidities [2]. In addition, the interaction of immune cells and the extracellular matrix (ECM) also plays an important role in the onset of low-grade inflammation in obesity. This is driven by the release of different molecules, such as small fragments of hyaluronan, which induce the activation of innate immune cells via toll-like receptors. Moreover, the involvement of these molecules in the development of insulin resistance has previously been suggested [31].

The present study brings new light in regard to the phenotype and function of circulating immune cells in people with obesity, with obesity-associated insulin resistance and T2D. The study aimed to obtain a deep characterization of circulating T cells. The impact of obesity on the imbalance of immune homeostasis is evident in both innate and adaptive immune systems. Higher white blood cell counts were observed in people with obesity, except for those in the insulin-sensitive group who displayed similar WBC counts as the nOB. Increased percentages of neutrophils and a reduction in T lymphocytes, with consequent reduction in total lymphocytes, in people with obesity were also observed. In previous studies, the important role of neutrophils was reflected in the onset of the inflammatory cascade in obesity [32] and in the onset of insulin resistance by the release of elastase, promoting the degradation of insulin receptors [7,33]. Moreover, different authors indicated the impact of obesity on the adaptive immune cells, with emphasis on T cells, either in circulation or infiltrated in adipose tissue [11]. In our cohort, increases in CD4^+^ T cells, accompanied by reductions in CD8^+^ T cells, were observed within circulating immune cells. 

In 2015, Łuczyński and collaborators found an increased count of circulating CD4^+^ T cells with the Th17 phenotype in children with obesity (10–18 years old) and suggested these cells played a key role in low-grade inflammation and established the link between obesity and diabetes [34]. In addition, they also observed a higher percentage of Th17 cells expressing IFN-γ [34] that have been further identified as Th1/17 cells [35]. Importantly, our data showed that both populations were increased in the peripheral blood of people with obesity, accompanied by a reduction in CCR5^−^CCR6^−^ Th cells. This profile was observed independently of their metabolic status. However, despite no difference being observed between the Th1 cell percentages, the IS and IRn participants presented lower expression of PD-1 by Th1 cells, as compared to T2D participants. The reduction in the expression of PD-1 could be indicative of an increased function of these cells in those participants. According to previous studies [reviewed [36]], the expression of PD-1 by Th cells was related to loss of function [36] and exhaustion [37]. In fact, studies indicated that the blockade of PD-1, as well as the blockade of TIM-3, induced an increase in IFN-γ production by PD-1^+^CD4^+^ T cells [36]. On the other hand, CD8^+^ T and CD4^+^CD8^+^ T cells from the OB group showed a different polarization tendency, with preferential Tc1- and T1-like polarization, respectively. Noteworthy, the Tc1 cells from individuals with obesity showed higher expression of CCR5, allowing these cells to migrate into inflammatory sites [38].

Some controversy remains regarding the circulating CD4^+^ Treg cells in people with obesity. Some studies indicate that CD4^+^ Treg cells are reduced in circulation in people with obesity [14], while others do not [39]. In the present study, no differences were identified in the percentage of CD4^+^ Treg cells within CD4^+^ T cells, although a significant reduction was observed in the percentage of CD4^+^ Treg cells measured within whole blood. It is important to mention that this reduction can be attributed to the increased white blood cell count in people with obesity, rather than to an actual reduction in the number of CD4^+^ Tregs in the PB. Therefore, we evaluated the absolute number of CD4^+^ Tregs in whole blood and found that the absolute count was not significantly different between the nOB and the OB groups, which displayed similar counts (Appendix A). Interestingly, CD8^+^ Treg cells follow the opposite direction, displaying an increase in their percentage in people with obesity—more marked in people with pre-diabetes [30]. It is noteworthy that this cell population was previously described as being positively correlated to BMI [40]. The expression of chemokine receptors by Treg cells allows them to migrate into inflammatory locations [38]. As previously reported, CD4^+^ T cells infiltrated into adipose tissue display a higher tendency to polarize toward Th1 cells [13] (and our unpublished data). It is important to acknowledge that the increase in the percentage of circulating Th1- and Th1/17-like Treg cells in people with obesity described in our study may indicate active migration of these Treg cells into adipose tissue as an attempt to mitigate the inflammation in the tissue microenvironment. Additionally, this increase was accompanied by a worsened metabolic condition. Furthermore, regarding the expression of the immune checkpoint proteins, PD-1 has commonly been associated with Treg homeostasis and immunosuppressive capability [reviewed [41]], as well as TIM-3 [42]. In fact, people with obesity and pre-diabetes showed a reduced percentage of PD-1^+^Th1-like Treg in circulation but also a reduction in PD-1^+^CD8^+^ Treg cells and TIM-3^+^CD8^+^ Treg cells [30]. This fact supports a previous study reporting the impaired immune suppressive function of Tregs during insulin resistance and T2D [43].

Follicular T cells play an important role in the maintenance and differentiation of B cells [44]. Guo and collaborators previously reported that people with T2D display higher percentages of Tfh cells in circulation, in agreement with our study. It is important to mention that the research conducted by Guo and collaborators (2020) was performed in people without obesity (BMI of 25.90 ±.99) [44]. Regardless of obesity, T2D can induce alterations in the percentage of Tfh cells. Interestingly, Tfc displayed a similar tendency, despite the expression of CXCR5 being reduced in T2D compared to the IS group. 

People with Pre-T2D and T2D displayed a higher expression of TIM-3 in some CD4^+^ T and CD8^+^ T cell subsets, compared to the remaining OB groups. However, Sun and collaborators (2020) reported a reduction in the expression of TIM-3 within CD4^+^ and CD8^+^ T cells, when comparing people with obesity and people with obesity and T2D [45]. In fact, the authors attributed the difference in the expression of TIM-3 to a transient upregulation of plasma glucose levels during the early stages of T2D, with a restoration of the T cells function in long-term T2D. Moreover, an increase in TIM-3 has also been previously described in Tfh cells [44].

Importantly, bariatric surgery has been used as a gold standard in the treatment of obesity and, consequently, obesity-related comorbidities [15]. The impact of this procedure has been well described regarding its positive effects on metabolic profiles. However, the effects of bariatric surgery have not been well studied regarding immunometabolism and the modulation of the immune system. Here, we present a deep characterization of T cells before and post-surgery. Our results show that bariatric surgery has important modulatory effects on the circulatory immune profile of people with obesity. Wijngaarden et al. (2022) postulated a recovery of the CD4^+^ T cell compartment in patients three months after bariatric surgery toward the lean control phenotype and found no changes in CD8^+^ T cells [15]. Our study analyzes the circulating immune profile of people with obesity, 7 to 18 months post-bariatric surgery. A higher percentage of CD4^+^ Tregs and B cells within whole blood was observed, with an increased activation profile, also accompanied by an expansion of Tfh.

A large heterogeneity found among participants and the medication used to treat/mitigate the obesity comorbidities may interfere directly or indirectly with the immune system. The most prescribed anti-diabetic, metformin, greatly impacts immune modulation, as reviewed in [46]. Furthermore, statins also play an important role in immune regulation by inhibiting T cell proliferation [47].

Despite these confounding factors, we demonstrated important new alterations on several T cell subsets associated with obesity. How distinct immune profiles impact the development of comorbidities and the obesity itself emerges as an interesting research area. Similarly, it is important to understand whether the outcome of bariatric surgery differs for patients with different immune profiles. Finally, our study demonstrated that obesity modulates differently distinct T cell subsets; therefore, it is mandatory to accurately identify these T cell subsets to understand which ones are altered. Otherwise, some differences observed in obesity would go unnoticed.

## 5. Conclusions

We describe significant and novel changes in circulating T cells associated with obesity. Remarkably, CD4^+^ and CD8^+^ T cells exhibited distinct alterations, even though both subsets exhibited increased CCR5 expression, potentially serving as a common link between the two populations. Notably, T cells from individuals with obesity displayed a pro-inflammatory profile and presented changes in the expression of immune regulatory molecules, which may serve as a counterbalance to mitigate the inflammation exacerbation. The precise impact of the immune alterations reported in this study on the development of insulin resistance and type 2 diabetes remains uncertain. The effects of bariatric surgery on the immune system were observed around 12 months after surgery, but it is mandatory to follow the participants for a longer period to understand the long-term effects of bariatric surgery on the different immune cells.

## Figures and Tables

**Figure 1 biomolecules-14-00219-f001:**
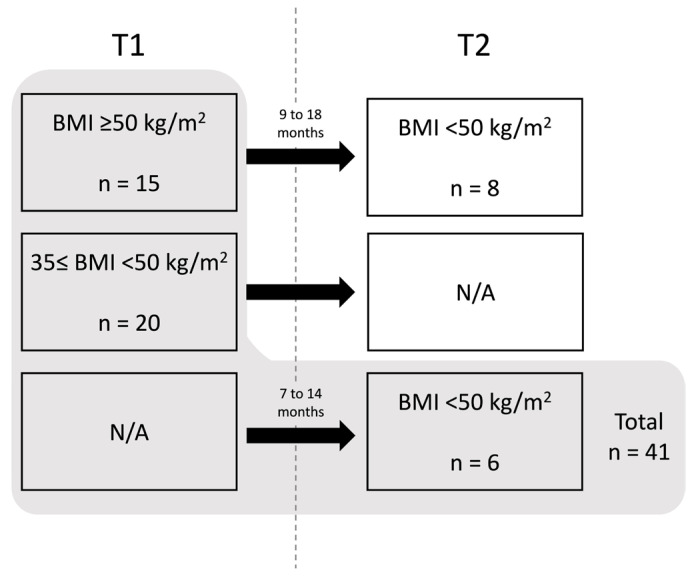
Distribution of the participants according to the time-point of sample collection. Thirty-five individuals were studied before bariatric surgery (T1). Eight of them, with BMI ≥50 kg/m^2^ at T1, were also studied 9 to 18 months after the surgery (T2). Six individuals were only studied at T2. N/A: not analyzed.

**Figure 2 biomolecules-14-00219-f002:**
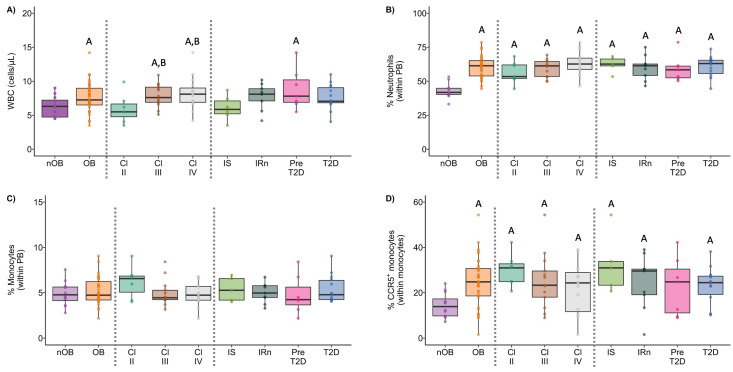
White blood cells count (WBC) in peripheral blood (**A**) and percentage of neutrophils (**B**) and monocytes within peripheral blood (PB) leukocytes (**C**) in nOB and OB, as well as among OB participants stratified by obesity class and metabolic profile. Percentage of monocytes expressing CCR5^+^ within all the studied groups (**D**). nOB: healthy participants (without obesity); OB: participants with obesity; IS: insulin sensitive; IRn: insulin resistant and normoglycemic; Pre-T2D: pre-diabetes; T2D: type 2 diabetes. Statistical differences were considered when *p* < 0.05. ^A^ *p* < 0.05 vs. nOB; ^B^ *p* < 0.05 vs. Class II; ^C^ *p* < 0.05 vs. Class III; ^D^ *p* < 0.05 vs. IS; ^E^ *p* < 0.05 vs. IRn; ^F^ *p* < 0.05 vs. Pre-T2D.

**Figure 3 biomolecules-14-00219-f003:**
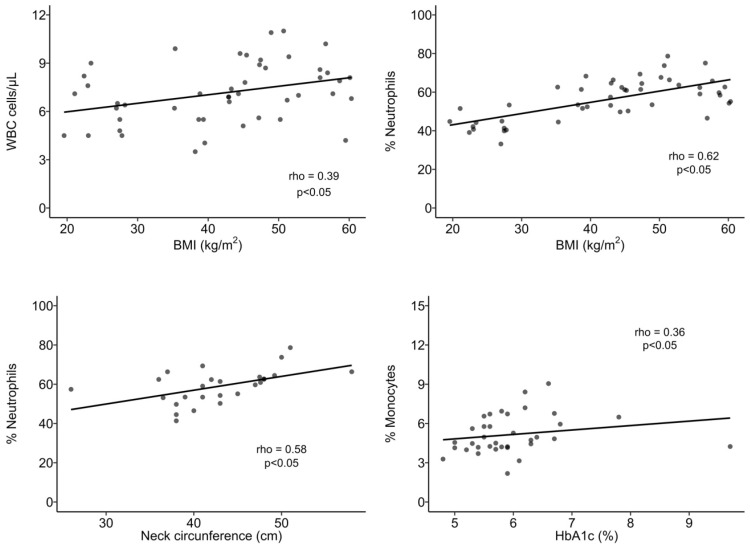
Spearman’s correlation analysis between immune and anthropometric or metabolic parameters. BMI: body mass index; WBC: white blood cells. Statistical differences were considered when *p* < 0.05.

**Figure 4 biomolecules-14-00219-f004:**
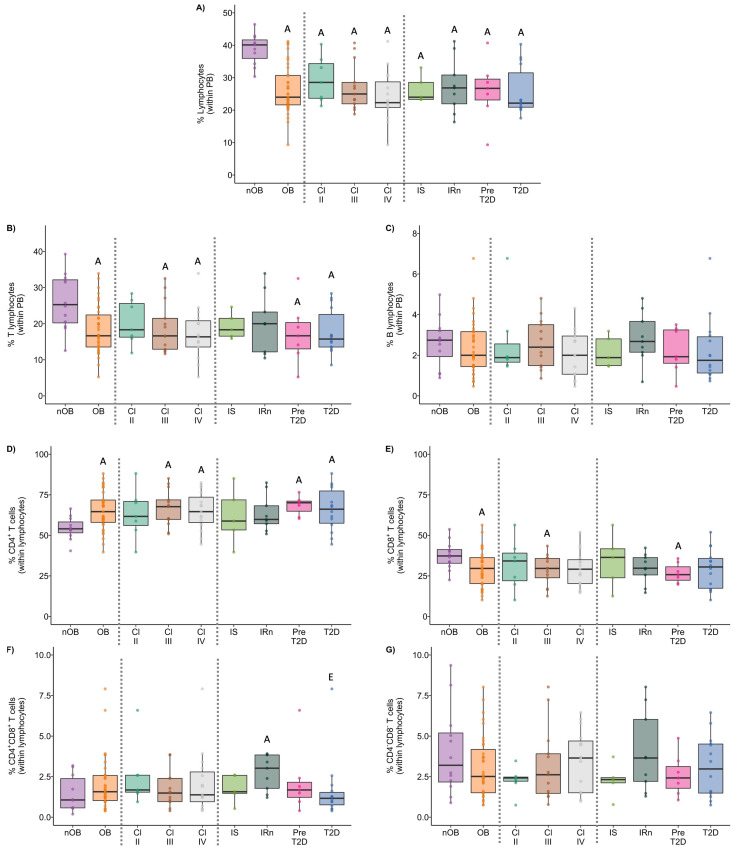
Percentage of lymphocytes (**A**), T lymphocytes (**B**), and B lymphocytes (**C**) within peripheral blood (PB) leukocytes and distribution of T lymphocytes into CD4^+^ (**D**), CD8^+^ (**E**), CD4^+^CD8^+^ (**F**), and CD4^−^CD8^−^ (**G**) compartments among all the studied groups. nOB: healthy participants (without obesity); OB: participants with obesity; IS: insulin sensitive; IRn: insulin resistant and normoglycemic; Pre-T2D: pre-diabetes; T2D: type 2 diabetes. Statistical differences were considered when *p* < 0.05. ^A^ *p* < 0.05 vs. nOB; ^B^ *p* < 0.05 vs. Class II; ^C^ *p* < 0.05 vs. Class III; ^D^ *p* < 0.05 vs. IS; ^E^ *p* < 0.05 vs. IRn; ^F^ *p* < 0.05 vs. Pre-T2D.

**Figure 5 biomolecules-14-00219-f005:**
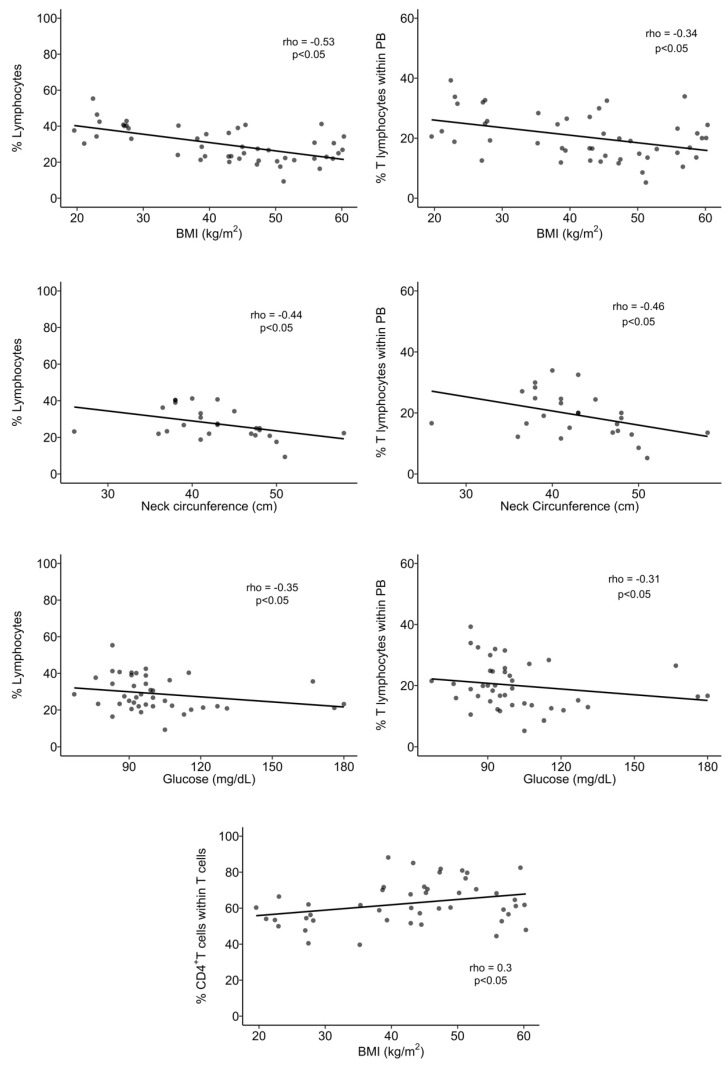
Spearman’s correlation analysis between immune and anthropometric or metabolic parameters. BMI: body mass index; PB: peripheral blood. Statistical differences were considered when *p* < 0.05.

**Figure 6 biomolecules-14-00219-f006:**
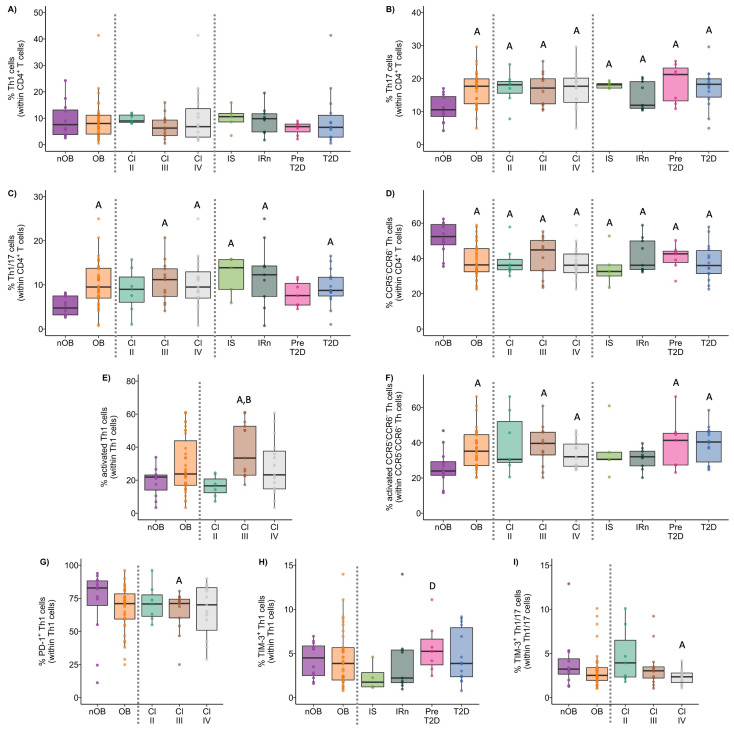
Percentage of Th1 (**A**), Th17 (**B**), Th1/17 (**C**), and CCR5^−^CCR6^−^ (**D**) T cells within CD4^+^ T cells. Percentage of activated (CD25^+^) Th1 (**E**) and CCR5^+^CCR6^+^ Th cells (**F**). Percentage of PD-1^+^ Th1 (**G**), TIM-3^+^ Th1 (**H**), and TIM-3^+^ Th1/17 cells (**I**) in nOB and OB, grouped according to obesity class and metabolic profile. nOB: healthy participants (without obesity); OB: participants with obesity; IS: insulin sensitive; IRn: insulin resistant and normoglycemic; Pre-T2D: pre-diabetes; T2D: type 2 diabetes; Th1—T helper 1 (CCR5^+^CCR6^−^); Th17—T helper 17 (CCR5^−^CCR6^+^); Th1/17—T helper 1/17 (CCR5^+^CCR6^+^). Statistical differences were considered when *p* < 0.05. ^A^ *p* < 0.05 vs. nOB; ^B^ *p* < 0.05 vs. Class II; ^C^ *p* < 0.05 vs. Class III; ^D^ *p* < 0.05 vs. IS; ^E^ *p* < 0.05 vs. IRn; ^F^ *p* < 0.05 vs. Pre-T2D.

**Figure 7 biomolecules-14-00219-f007:**
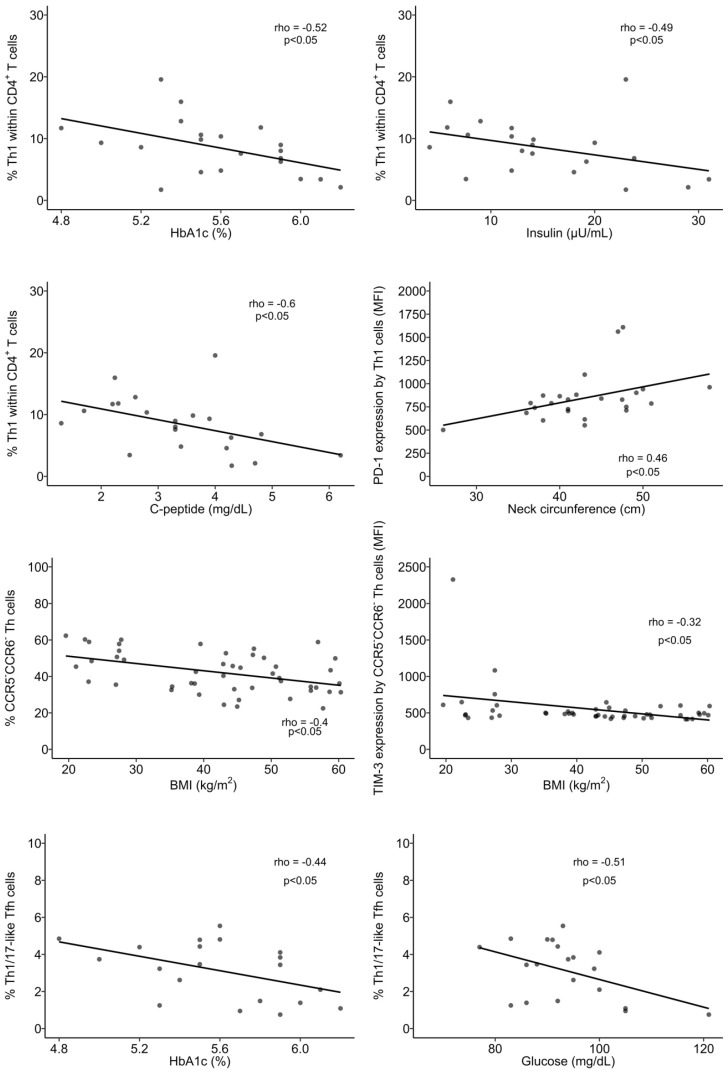
Spearman’s correlation analysis between immune and anthropometric or metabolic parameters. BMI: body mass index; MFI: mean fluorescence intensity. Statistical differences were considered when *p* < 0.05.

**Figure 8 biomolecules-14-00219-f008:**
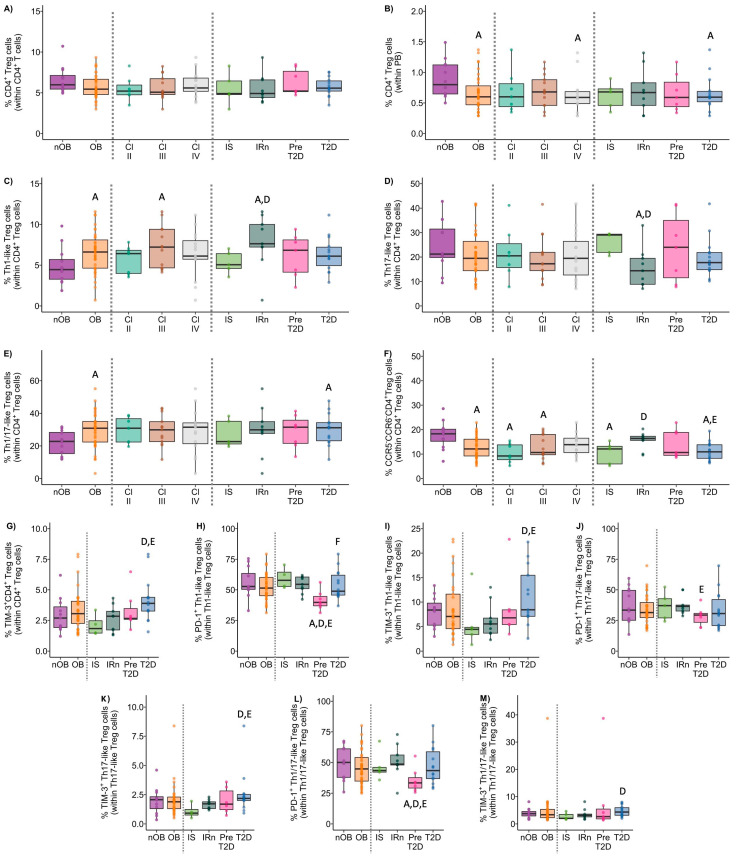
Percentage of CD4^+^ Tregs within CD4^+^ T cells (**A**) and within peripheral blood (PB) leukocytes (**B**). Polarization of CD4^+^ Tregs into Th1-like (**C**), Th17-like (**D**), Th1/17-like (**E**), and CCR5^−^CCR6^−^ (**F**) CD4^+^ Tregs. Percentage of PD-1^+^ and TIM-3^+^ expressed by CD4^+^ Tregs (**G**) and by Th1-like (**H**,**I**), Th17-like (**J**,**L**), and Th1/17-like (**M**,**N**) CD4^+^ Tregs among the studied groups. nOB: healthy participants (without obesity); OB: participants with obesity; IS: insulin sensitive; IRn: insulin resistant and normoglycemic; Pre-T2D: pre-diabetes; T2D: type 2 diabetes. Statistical differences were considered when *p* < 0.05. ^A^ *p* < 0.05 vs. nOB; ^B^ *p* < 0.05 vs. Class II; ^C^ *p* < 0.05 vs. Class III; ^D^ *p* < 0.05 vs. IS; ^E^ *p* < 0.05 vs. IRn; ^F^ *p* < 0.05 vs. Pre-T2D.

**Figure 9 biomolecules-14-00219-f009:**
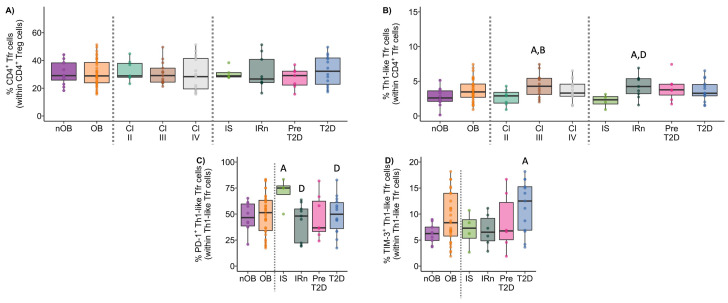
Treg cells with follicular phenotype and their polarization into Th1-like Tfr cells. Percentage of follicular CD4^+^ Treg cells (Tfr) within CD4^+^ Treg cells (**A**). Percentage of CD4^+^ Tfr cells with Th1-like phenotype (**B**). Percentage of PD-1^+^ (**C**) and TIM-3^+^ (**D**) cells within Th1-like CD4^+^ Tfr cells. nOB: healthy participants (without obesity); OB: participants with obesity; IS: insulin sensitive; IRn: insulin resistant and normoglycemic; Pre-T2D: pre-diabetes; T2D: type 2 diabetes. Statistical differences were considered when *p* < 0.05. ^A^ *p* < 0.05 vs. nOB; ^B^ *p* < 0.05 vs. Class II; ^C^ *p* < 0.05 vs. Class III; ^D^ *p* < 0.05 vs. IS; ^E^ *p* < 0.05 vs. IRn; ^F^ *p* < 0.05 vs. Pre-T2D.

**Figure 10 biomolecules-14-00219-f010:**
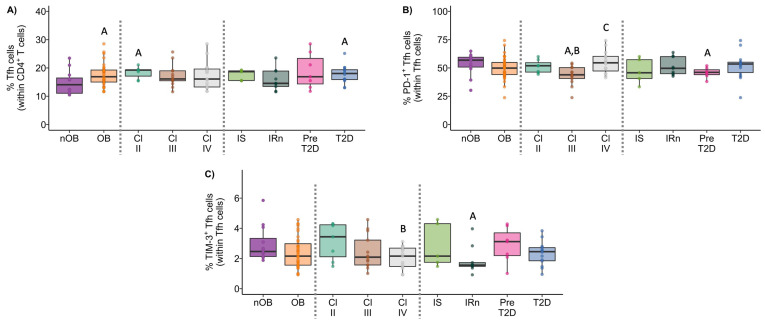
Frequency and expression of immune regulatory molecules by follicular CD4^+^ T cells (Tfh). Percentage of follicular CD4^+^ T cells (Tfh) within CD4^+^ T cells (**A**). Percentage of PD-1^+^ (**B**) and TIM-3^+^ (**C**) cells within Tfh cells, in the different studied groups. nOB: healthy participants (without obesity); OB: participants with obesity; IS: insulin sensitive; IRn: insulin resistant and normoglycemic; Pre-T2D: pre-diabetes; T2D: type 2 diabetes. Statistical differences were considered when *p* < 0.05. ^A^ *p* < 0.05 vs. nOB; ^B^ *p* < 0.05 vs. Class II; ^C^ *p* < 0.05 vs. Class III; ^D^ *p* < 0.05 vs. IS; ^E^ *p* < 0.05 vs. IRn; ^F^ *p* < 0.05 vs. Pre-T2D.

**Figure 11 biomolecules-14-00219-f011:**
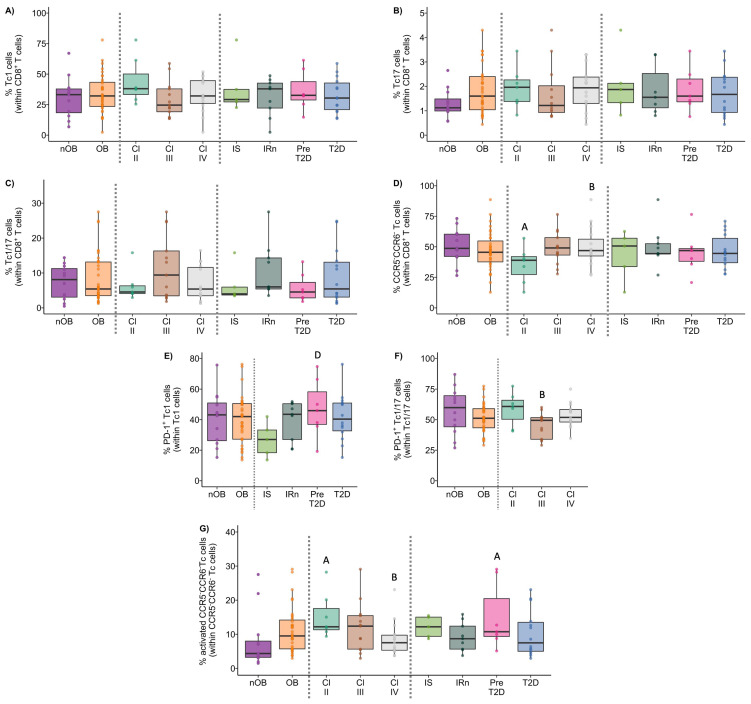
The Tc1 (**A**), Tc17 (**B**), Tc1/17 (**C**), and CCR5^−^CCR6^−^ (**D**) cell percentages within CD8^+^ T cells, as well as the percentages of PD-1^+^ Tc1 (**E**) and Tc1/17 (**F**) cells and of the activated CCR5^−^CCR6^−^ Tc cells (**G**) in nOB and OB, grouped according to obesity class and metabolic profile. nOB: healthy participants (without obesity); OB: participants with obesity; IS: insulin sensitive; IRn: insulin resistant and normoglycemic; Pre-T2D: pre-diabetes; T2D: type 2 diabetes; Tc1—T cytotoxic 1 (CCR5^+^CCR6^−^); Tc17—T cytotoxic 17 (CCR5^−^CCR6^+^); Tc1/17—T cytotoxic 1/17 (CCR5^+^CCR6^+^). Statistical differences were considered when *p* < 0.05. ^A^ *p* < 0.05 vs. nOB; ^B^ *p* < 0.05 vs. Class II; ^C^ *p* < 0.05 vs. Class III; ^D^ *p* < 0.05 vs. IS; ^E^ *p* < 0.05 vs. IRn; ^F^ *p* < 0.05 vs. Pre-T2D.

**Figure 12 biomolecules-14-00219-f012:**
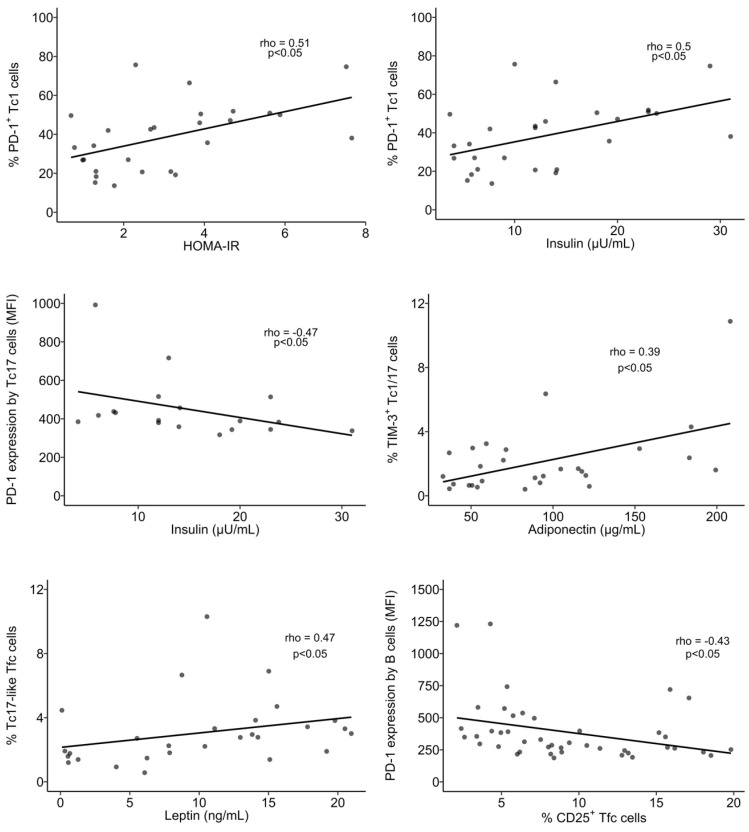
Spearman’s correlation analysis between immune and anthropometric or metabolic parameters. BMI: body mass index; MFI: mean fluorescence intensity. Statistical differences were considered when *p* < 0.05.

**Figure 13 biomolecules-14-00219-f013:**
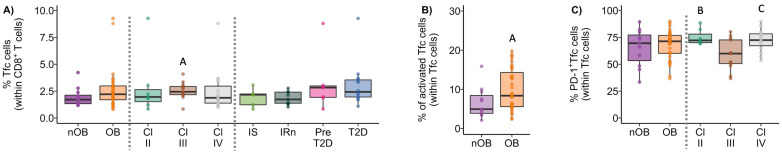
Frequency, activation profile, and immune regulatory markers expression by follicular CD8^+^ T cells (Tfc). Percentage of follicular CD8^+^ T cells (Tfc) within CD8^+^ T cells (**A**). Percentage of activated Tfc (**B**) and PD-1^+^ Tfc (**C**) cells in the different groups under study. nOB: healthy participants (without obesity); OB: participants with obesity; IS: insulin sensitive; Irn: insulin resistant and normoglycemic; Pre-T2D: pre-diabetes; T2D: type 2 diabetes. Statistical differences were considered when *p* < 0.05. ^A^ *p* < 0.05 vs. nOB; ^B^ *p* < 0.05 vs. Class II; ^C^ *p* < 0.05 vs. Class III; ^D^ *p* < 0.05 vs. IS; ^E^ *p* < 0.05 vs. IRn; ^F^ *p* < 0.05 vs. Pre-T2D.

**Figure 14 biomolecules-14-00219-f014:**
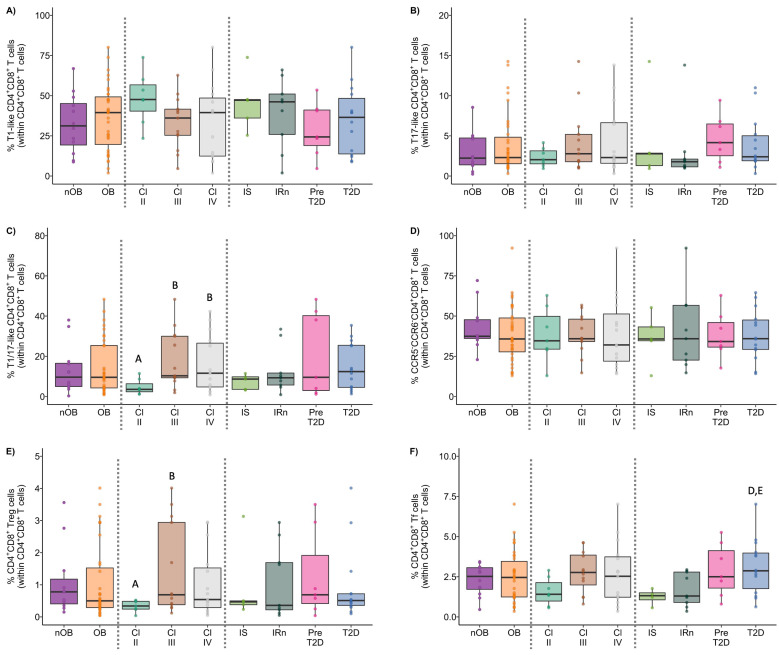
Percentage of CD4^+^CD8^+^ T cells presenting a T1-like (**A**), T17-like (**B**), T1/17-like (**C**), and CCR5^−^CCR6^−^ (**D**), Treg (**E**), and Tf (**F**) phenotype, within CD4^+^CD8^+^ T cells, in nOB and OB, grouped according to obesity class and metabolic profile. nOB: healthy participants (without obesity); OB: participants with obesity; IS: insulin sensitive; IRn: insulin resistant and normoglycemic; Pre-T2D: pre-diabetes; T2D: type 2 diabetes; T1-like—CCR5^+^CCR6^−^; T17-like—CCR5^−^CCR6^+^; T1/17-like—CCR5^+^CCR6^+^. Statistical differences were considered when *p* < 0.05. ^A^ *p* < 0.05 vs. nOB; ^B^ *p* < 0.05 vs. Class II; ^C^ *p* < 0.05 vs. Class III; ^D^ *p* < 0.05 vs. IS; ^E^ *p* < 0.05 vs. IRn; ^F^ *p* < 0.05 vs. Pre-T2D.

**Figure 15 biomolecules-14-00219-f015:**
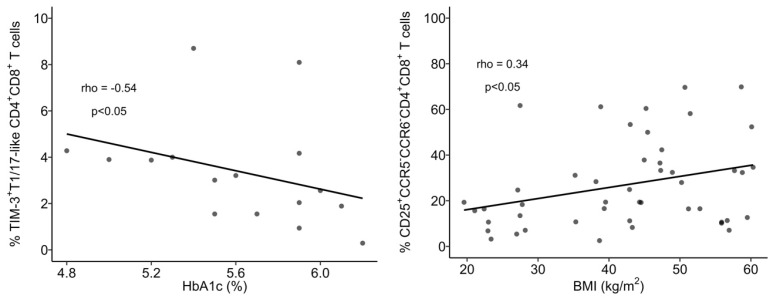
Spearman’s correlation analysis between immune and anthropometric or metabolic parameters. BMI: body mass index. Statistical differences were considered when *p* < 0.05.

**Figure 16 biomolecules-14-00219-f016:**
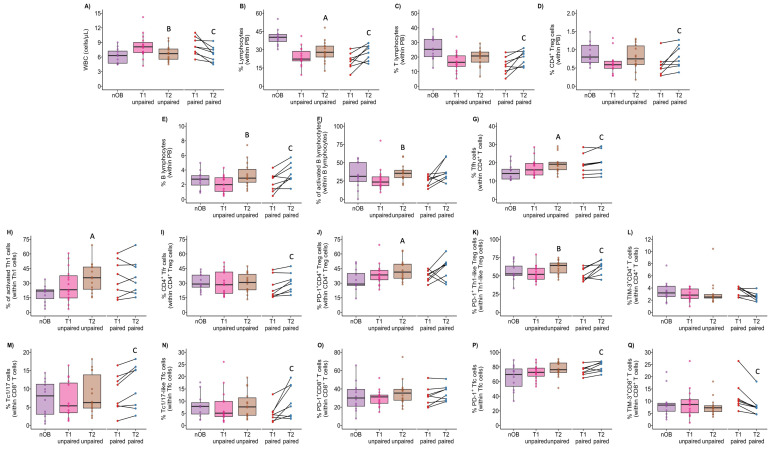
Phenotypical changes in T cells from individuals with obesity, before (T1) and after (T2) bariatric surgery. The analysis of unpaired samples comprised all individuals with Class IV obesity in T1 (*n* = 15) and 14 individuals after bariatric surgery (T2). The analysis of paired samples consisted in a follow-up of 8 individuals before (T1) and after (T2) bariatric surgery. The nOB group was used as reference. Distribution of white blood cells count (WBC) (cells/µL) before (T1) and after (T2) surgery (**A**). Percentage of lymphocytes (**B**), T lymphocytes (**C**), CD4^+^ Treg cells (**D**), and B lymphocytes (**E**), within peripheral blood leukocytes (PB), before (T1) and after (T2) surgery. Percentage of activated B cells (measured within B cells) (**F**) and Tfh cells (within CD4^+^ T cells) (**G**), activated Th1 cells (within Th1 cells) (**H**), and CD4^+^ Tfr cells (within CD4^+^ Treg cells) (**I**) before (T1) and after (T2) surgery. Percentages of PD-1^+^ CD4^+^ Tregs (**J**) and Th1-like Tregs (**K**) and percentage of TIM-3^+^CD4^+^ T cells (**L**) before (T1) and after (T2) bariatric surgery. Percentage of Tc1/17, within CD8^+^ T cells (**M**) and Tc1/17-like Tfc cells, within Tfc (**N**), at T1 and T2. Percentages of PD-1^+^ CD8^+^ T cells (**O**) and Tfc cells (**P**) and percentage of TIM-3^+^CD8^+^ T cells (**Q**) before (T1) and after (T2) surgery. nOB: healthy participants (without obesity). Statistical differences were considered when *p* < 0.05. ^A^ *p* < 0.05 vs. nOB; ^B^ *p* < 0.05 vs. T1 unpaired; ^C^ *p* < 0.05 vs. T1 paired.

**Table 1 biomolecules-14-00219-t001:** Anthropometric and biochemical characterization of participants, before (T1) and after (T2) bariatric surgery.

	**nOB**		**OB at T1**		**OB at T2**	
	*n*		*n*		*n*	
Sex (female/male)	12	8/4	35	23/12	14	8/6
Age (years)	12	43 ± 12	35	45 ± 12	14	41 ± 9.0
BMI (kg/m^2^)	12	24.8 ± 3.0	35	48.3 ± 7.6	14	41.6 ± 4.3 **^A,B^**
NC (cm)			25	43 ± 6	13	41 ± 5 **^B^**
WHR	6	0.82 ± 0.11	25	0.98 ± 0.10 **^A^**	13	0.96 ± 0.10 **^A^**
Obesity class (II/III/IV)			35	7/13/15	14	6/8/-
Type 2 Diabetes			35	14	14	2
HbA1c (%)	5	5.5 ± 0.48	35	6.0 ± 0.88	14	5.1 ± 0.21 **^A,B^**
Glucose (mg/dL)	7	88.6 ± 8.0	35	104.8 ± 25.5	14	83.2 ± 6.3 **^B^**
Insulin (µU/mL)	6	5.9 ± 2.3	35	17.1 ± 9.1 **^A^**	14	6.1 ± 2.0 **^B^**
HOMA-IR	6	1.3 ± 0.54	35	4.6 ± 3.2 **^A^**	14	1.5 ± 0.76 **^B^**
C-peptide (ng/mL)	3	1.34 ± 0.43	35	4.0 ± 1.5 **^A^**	13	2.1 ± 0.57 **^B^**
LDL (mg/dL)	7	101.7 ± 32.5	35	113.2 ± 41.2	14	118.7 ± 31.5
HDL (mg/dL)	7	67.0 ± 19.3	35	43.8 ± 10.9 **^A^**	14	44.9 ± 14.6 **^A^**
Triglycerides (mg/dL)	7	101.3 ± 52.7	35	124.8 ± 52.1	14	84.6 ± 26.5 **^B^**
Total cholesterol (mg/dL)	7	189.0 ± 53.2	35	182.5 ± 40.3	14	180.6 ± 36.4
Atherogenic index	6	3.0 ± 0.61	35	4.5 ± 1.4 **^A^**	14	4.4 ± 1.2 **^A^**
CRP (mg/dL)	3	0.20 ± 0.23	35	1.0 ± 0.77 **^A^**	14	0.64 ± 0.52 **^B^**
Leptin (ng/mL)	6	2.6 ± 3.5	22	12.0 ± 6.0 **^A^**	10	5.9 ± 5.2 **^B^**
Adiponectin (µg/mL)	6	121.1 ± 57.9	22	86.2 ± 48.7	10	138.6 ± 121.9

nOB: healthy participants (without obesity); OB at T1: group of all individuals with obesity studied before surgery; OB at T2: group of all individuals studied 7 to 18 months after bariatric surgery; BMI: body mass index; NC: neck circumference; WHR: waist-to-hip ratio; HOMA-IR: homeostatic model assessment of insulin resistance; LDL: low-density lipoprotein cholesterol; HDL: high-density lipoprotein cholesterol; CRP: C-reactive protein. Statistical differences were considered when *p* < 0.05. ^A^ *p* < 0.05 vs. nOB; ^B^ *p* < 0.05 vs. OB before surgery, considering only participants with BMI ≥50 kg/m^2^ (obesity Class IV).

**Table 2 biomolecules-14-00219-t002:** Criteria used to classify people with obesity according to their metabolic profiles.

	**Insulin Sensitive (IS)**	**Insulin Resistant (IR)**		
HOMA-IR	<2	≥2		
		Insulin Resistant Normoglycemic (IRn)	Pre-Diabetes (Pre-T2D)	Type 2 Diabetes (T2D)
Fasting glucose (mg/dL)		<100	<125	≥125
HbA1c (%)		<5.7	≥5.70 and <6.49	≥6.5
	Non-Type 2 Diabetes (nT2D)			

**Table 3 biomolecules-14-00219-t003:** Distribution of participants with obesity according to their metabolic group and obesity class.

		**Metabolic Groups**			
		IS	IRn	Pre-T2	T2D
Obesity Class	Class II	3	0	2	2
	Class III	2	4	3	4
	Class IV	0	5	2	8
	Total	5	9	7	14

IS: insulin sensitive; IRn: insulin resistant and normoglycemic; Pre-T2D: pre-diabetes; T2D: type 2 diabetes.

**Table 4 biomolecules-14-00219-t004:** Monoclonal antibody combination used to identify the different T cell subsets, presenting the information on each respective fluorochrome, clone, and commercial source.

Monoclonal Antibodies											
	TIM-3	CD25	CD4	CD127	CD3	CD20	CD8	CXCR5	CCR6	PD-1	CCR5
Fluorochrome	FITC	PE	PerCP-Cy5.5	PE-Cy7	APC	APC-R700	APC-H7	BV421	BV510	BV605	BV711
Clone	7D3	M-A251	SK3	HIL-7R-M21	UCHT1	2H7	SK1	RF8B2	11A9	EH12.1	2D7/CCR5
Commercial source	BD Horizon	BD Pharmingen	BD Pharmingen	BD Pharmingen	BD Pharmingen	BD Horizon	BD Pharmingen	BD Horizon	BD Horizon	BD Horizon	BD Horizon

FITC: fluorescein isothiocyanate; PE: phycoerythrin; PerCP-Cy5.5: peridin chlorophyll protein–cyanine 5.5; PE-Cy7: phycoerythrin/cyanine 7; APC: allophycocyanin; APC-H7: allophycocyanin-Hilite7; APC-R700: allophycocyanin R700; BV: Brilliant Violet; BD Pharmingen (San Diego, CA, USA); BD Horizon (San Jose, CA, USA).

## Data Availability

Dataset available on request from the authors.

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
