# Peer review of "Bariatric Surgery Induces Alterations in the Immune Profile of Peripheral Blood T Cells"

_biomolecules, 2024, doi:10.3390/biom14020219_

Round 1
Reviewer 1 Report
Comments and Suggestions for Authors
First I want to thank you for your study. However, I found the results too difficult to assess. Flow data outputs should not be presented as just tables. The data should be transferred into a graph format, showing markers for patients and Mean +/- SD (Bar chart with individual plots). By presenting it as raw data it is almost impossible to understand the different conclusions.
I would suggest you choose the markers that showed the differences and present those graphs in the main document, and those that do not show any major differences to be transferred into supplementary figures. This would aid in the readability of the paper and allow for the pertinent data to be highlighted.
Reviewer 2 Report
Comments and Suggestions for Authors
The present study by Barbosa et al., brings new light in regard to the phenotype and function of circulating immune cells in people with obesity, with obesity-associated insulin resistance and T2D. The study presented aimed to obtain a deep characterization of circulating T cells.
Overall the study is well done and the data are well analyzed and presented.
Here some minor points of critique:
The distribution of participants is not clearly described, the text and figure 1 does not 100% align. Also, please clarify how the subset of patients for follow up analyzes were selected.
Table 1 does show that LDL is higher in the OB at T2 compared to OB at T1, what do the authors make of that? What would explain the increase in LDL after surgery?
The reviewer feels that figure 2, the Dotplot histograms illustrating the gating strategy would be better as a supplemental figure.
Obesity and immune profiles are influenced by the extracellular matrix (ECM), which has been published by many groups. Do the authors have any ECM marker they can include in their analysis? One of the most prominent ECM markers is hyaluronan. Hyaluronan for instance has been implicated in diabetes as well as in T cell and Treg changes. The authors should at least add a small paragraph and speculate about the influence of ECM on the seen changes after surgery.
Comments on the Quality of English Language
Minor editing of English language required.
Round 2
Reviewer 1 Report
Comments and Suggestions for Authors
Dear Authors,
Thank you. I must commend you on really listening to my previous comment and making the manuscript more readable. I think you should change the A,B,C,D when referring to statistics with the more conventional * or ** as it does get confusing when referring to multiple panels. Overall well done for this important study on T cells before and after bariatric surgery in obese patients. I can't wait for a follow-up study looking at other interventions such as weight loss drugs (Ozempic).
